# OpenReview forum: "Bitune: Leveraging Bidirectional Attention to Improve Decoder-Only LLMs"
_ICLR.cc/2025/Conference — Submitted to ICLR 2025_

### Official Review · Reviewer_3Yh8 · 2024-10-21

**Soundness:** 4
**Presentation:** 4
**Contribution:** 3
**Rating:** 8
**Confidence:** 4

**Summary:**

This paper presents “Bitune”, a method that enables pre-trained causal LMs (unidirectional attention), to use bidirectional attention. The proposed Bitune method makes use of sparse LoRA-based adaptors to create a bidirectional version of the model. Each prompt is passed through both the causal and bidirectional version of the model, and then both sets of features are combined into a single (key, value) cache. This cache is used to generate the rest of the sequence.

**Strengths:**

1. The paper does demonstrate a performance gain from Bitune compared to the baseline pre-trained model and LoRA fine-tuned model on several language modeling benchmarks (table 1,2).
2. Paper does a good job of doing a thorough ablation study to verify the necessity of different components of their proposed method in section 3.3.
3. Paper does discuss the computational cost of adapting the base model to be bi-directional and provides a reasonable solution to mitigate the computational overhead: using parameter efficient fine-tuning. They also highlight that Bitune is not tied to the specific method for updating weights, and instead can be adapted to use different types of weight updating methods.

**Weaknesses:**

Each round of Bitune inference effectively requires two inference passes which can be (time and memory) expensive (especially for really long sequence generation tasks). The authors do highlight this limitation in the discussion section and Appendix 6.4. According to their results, it seems that the most impacted part of the training/inference pipeline is the additional training time cost. Inference time costs did increase but to a much lesser extent. This timing increase could be reasonable if an end user of Bitune feels that the performance gains of the Bitune model warrants the extra computational time. I note that the time analysis is done for a fixed generation length of 200 tokens; I am curious to see how the time scales as generation length increases (e.g., 2000 tokens generated).

**Questions:**

n/a

---

> ### Author Response · Authors · 2024-11-22
>
> Thank you for your positive feedback!
>
> To answer the question on how the inference time scales with longer generations: Bitune affects only the prefilling part of the inference process, and there is **no overhead** for generation of new tokens. Therefore, the relative increase in time diminishes with the length of the generated sequence. Please see below for a table with runtimes at various output token lengths, for Gemma-2B model, and **long instruction** of length 2000, on A100 GPU:
>
>
> | Model   | TTFT  | 200 output tokens | 400 o.t. | 2000 o.t. | 4000 o.t. |
> |---------|-------------|------------------|----|----|----|
> | LoRA    | 0.03s  | 6.83s | 12.30s | 62.15s | 128.77s |
> | Bitune  | 0.28s  | 6.85s | 12.83s | 62.28s | 128.90s |
>
>
> We can see that _time to first token_ (TTFT), which consists only of the prefilling part, increases from 0.03s to 0.28s. However, it's a one-time cost, and the increase is insignificant when we consider the total time required to generate longer sequences.

---

> > ### Comment · Reviewer_3Yh8 · 2024-11-26
> >
> > Thank you for clarifying. As my initial score was already high, I will leave it as is.

---

### Official Review · Reviewer_B9De · 2024-11-02

**Soundness:** 3
**Presentation:** 3
**Contribution:** 2
**Rating:** 5
**Confidence:** 4

**Summary:**

The paper aims to improve decoder-only LLMs' effectiveness in downstream tasks by enhancing their representation of the input prompts. To achieve that, they introduce bi-directional attention into prompt processing. An LLM learns to represent a given prompt with both causal and bi-directional attention mechanisms and then fuse two features with the learnable combination. Experiments demonstrate the effectiveness of the proposed method in instruction-tuning and question-answering settings. Further ablation study verifies the performance gain comes from the bi-directional attention instead of more parameters or the fusion of two features.

**Strengths:**

1. The proposed idea addresses the innate limitation of decoder-only LLMs while preserving their generation efficiency.
2. The experiments are comprehensive, covering various LLMs and downstream tasks.
3. The ablation study helps identify the contribution of different design choices.

**Weaknesses:**

1. As indicated by the Related Work section, there are several prior works investigating the idea of bi-directional attention in decoder-only LMs. The novelty of this work is thus somewhat limited. It might be helpful if the authors could further discuss how their method stands out.
2. It is unclear why decoder-only LLMs would still face the limitation of representing language using causal attention given that they have undergone extensive pre-training and fine-tuning based on causal attention. Perhaps the authors can provide more evidence besides empirical results such as analysis of the hidden states in cases where the previous context indeed matters for the following one.
3. The proposed method only applies to the given prompt but not the generated response, which could also benefit from bi-directional attention. Also, it is not convincing that this method would be widely used in practice.

**Questions:**

How could this method be generalized to represent the generated tokens?

---

> ### Author Response · Authors · 2024-11-22
>
> Thank you for your time and feedback. We would like to clarify the raised concerns.
>
> **How does Bitune compare to prior work?**
>
> >As indicated by the Related Work section, there are several prior works investigating the idea of bi-directional attention in decoder-only LMs. The novelty of this work is thus somewhat limited. It might be helpful if the authors could further discuss how their method stands out.
>
> The prior works utilizing bi-directional attention in decoder-only LMs focused on *pretraining*, while the proposed method aims to effectively improve available decoder-only models by reintroducing the full attention and *finetuning them* either on downstream tasks (Section 3.2), or on instruction-tuning datasets (Section 3.1). As we demonstrate in the paper, this can be done efficiently, even with academic compute.
>
> The most similar technique from prior works is the “prefix-lm” objective/architecture, and it’s equivalent to our ablation variant called _naive bidir._ (Section 3.3). It applies bidirectional attention to the prefix/query while decoding the answer with causal attention. Our proposed method differs in the following ways:
>
> - We apply bidirectional *and* causal attention to the query, yielding two sets of features.
> - We use separate weights for obtaining these two sets of features.
> - We fuse both sets of features with learnable coefficients.
>
> With results in Table 3 we show that **the proposed method is superior to prefix-lm**/naive bidir. approach. We will make this differentiation more clear in the final paper.
>
> **Limitations of causal attention**
>
> >It is unclear why decoder-only LLMs would still face the limitation of representing language using causal attention given that they have undergone extensive pre-training and fine-tuning based on causal attention. Perhaps the authors can provide more evidence besides empirical results such as analysis of the hidden states in cases where the previous context indeed matters for the following one.
>
> The fundamental limitation of causal attention persists regardless of training because it's an architectural constraint that affects how tokens can be processed. Consider this illustrative example:
>
> _“What is x in the equation 6*x+7=42?"_
>
>
> With causal attention, when processing the first tokens, the model must sequentially generate _general_ representations for these tokens without being able to "see" the mathematical expression ahead.
>
> Bidirectional attention, in contrast, can immediately begin processing the mathematical task from the start, bringing more capacity to the model and allowing it to more optimally spend the compute early in the sequence.
>
> A similar phenomenon has been observed in [1], where text embeddings have been found to be less expressive when earlier tokens cannot attend to those occurring later in the sequence.
>
>
> **Generalization to generated tokens & practical use of Bitune**
>
> > [...] it is not convincing that this method would be widely used in practice.
>
> Instruction-following has become the predominant usage pattern for LLMs since ChatGPT. **Our method improves performance in this setting, making it broadly applicable.** While we primarily focus on single-turn interactions, which are crucial for applications like smart devices and voice assistants, our approach naturally extends to multi-turn scenarios (see answer to the next question).
>
> >How could this method be generalized to represent the generated tokens?
>
> The proposed method can be also applied to features of already generated tokens by recomputing the KV cache. In chatbot applications, the most straightforward way would be to treat the whole chat history as a prefix/query, and recompute the KV-cache for every new round - this would leverage bidirectional attention on all tokens, including the generated answers in previous rounds.
>
>
> ---
>
> Thank you again for your feedback! We hope that our answer addresses your concerns and that you will consider raising your score.
>
> ---
>
> References
>
> [1] Repetition Improves Language Model Embeddings. Springer et al. 2024 ArXiv

---

> > ### Comment · Reviewer_B9De · 2024-11-25
> >
> > Thanks for the response!
> > - Given that what Bitune differs from prefix-lm is just fusing two sets of features (causal&bi-directional) instead of proposing causal attention, I think this further limits the novelty of the work.
> > - What is more concerning is that if the fundamental limitation of causal attention does exist, shouldn't we just use the bi-directional feature? What is the benefit of adding the causal feature?
> > - The need for recomputing kv-cache in multi-turn tasks is again what would hinder this approach to be broadly used in practice. In that case, why not go back to use the encoder-decoder architecture?

---

> > > ### Author Response · Authors · 2024-11-25
> > >
> > > Thanks for the response.
> > >
> > > >Given that what Bitune differs from prefix-lm is just fusing two sets of features (causal&bi-directional) instead of proposing causal attention, I think this further limits the novelty of the work.
> > >
> > > To the best of our knowledge, this work is the first to effectively turn existing decoder-only LMs into hybrid models with bidirectional attention in the QA/instruction-tuning setting, leading to significant improvements across different benchmarks. Moreover, the method itself is novel, as it exploits the inherent structure of this setting to introduce separate weights for obtaining two kinds of features, and fuses them with learnable coefficient.
> > >
> > > Next to the novelty, the impact of our work should also be factored in. With the proposed method, the most common LLM use-cases - instruction-following and QA - can be improved by significant margins without incurring significant overhead.
> > >
> > > >What is more concerning is that if the fundamental limitation of causal attention does exist, shouldn't we just use the bi-directional feature? What is the benefit of adding the causal feature?
> > >
> > > Causal attention has it's own benefits, for example it allows for creation of implicit positional encodings - explored in e.g. [1]. Additionally, the models were pretrained to use the causal attention, and simply removing the causal mask does not lead to the same gains. Fusing both types of features allows the model to leverage existing pretrained features, and new benefits introduced with bidirectional attention. We have verified that with an ablation study (naive bidir., Table 3).
> > >
> > > >The need for recomputing kv-cache in multi-turn tasks is again what would hinder this approach to be broadly used in practice.
> > >
> > > We have presented this solution as the question was specifically about _representing generated tokens with this method_. Another solution exists, with alternating attention pattern, that does not require recomputing the KV-cache:
> > >
> > > 1. Prefill the first instruction and generate initial KV-cache (by merging causal and bidirectional features).
> > > 2. Generate output token by token, appending KV of the output to the KV cache (here the KV of the output has only causal features).
> > > 3. Prefill the next instruction with two attention masks and append KV to the cache.
> > > 4. Repeat from step 2.
> > >
> > > Moreover, the cost of prefilling/KV-cache recomputation is negligible when compared to total time required to generate a sequence. Please see the below table with example generation of up to 4000 tokens for 2000 token instruction with _Bituned_ Gemma-2B:
> > >
> > > | Model   | TTFT  | 200 output tokens | 400 o.t. | 2000 o.t. | 4000 o.t. |
> > > |---------|-------------|------------------|----|----|----|
> > > | LoRA    | 0.03s  | 6.83s | 12.30s | 62.15s | 128.77s |
> > > | Bitune  | 0.28s  | 6.85s | 12.83s | 62.28s | 128.90s |
> > >
> > > (TTFT - time to first token)
> > >
> > > We can see that (unoptimized) fusing procedure leads to roughly 0.25s overhead when prefilling an instruction, which is a negligible fraction of total runtime.
> > >
> > > >In that case, why not go back to use the encoder-decoder architecture?
> > >
> > > That's a question out of scope for this work, but it has been shown that encoder-decoder models and prefix-LMs are indeed superior to popular causal decoder-only models [2][3]. Nevertheless, decoder-only models remain the most popular and capable ones, and with this work we aim to improve their performace further.
> > >
> > > ---
> > >
> > > [1] The Impact of Positional Encoding on Length Generalization in Transformers. Kazemnejad et al. 2023
> > >
> > > [2] Exploring the Limits of Transfer Learning with a Unified Text-to-Text Transformer. Raffel et al. 2019
> > >
> > > [3] UL2: Unifying Language Learning Paradigms. Tay et al. 2022
> > >
> > > ---
> > >
> > > We hope this response addresses your concerns. If you have any remaining questions, we're happy to clarify.

---

> > > ### Author Response · Authors · 2024-11-30
> > >
> > > As the discussion period is coming to a close, we would appreciate knowing if you have any remaining concerns about our work. If not, we kindly ask you to consider revising your score based on our clarifications.

---

### Official Review · Reviewer_e3Cz · 2024-11-04

**Soundness:** 4
**Presentation:** 4
**Contribution:** 4
**Rating:** 10
**Confidence:** 4

**Summary:**

This paper looks at how to add bidriectional attention to pre-trained decoder-only LLMs. This is done by learning new Key and Value matrices used during attention without the causal masking. Instead of just replacing the original matrices, they run the "encoder" part of the network (the parts with bidirectional attention) twice, once with the original weights and causal masking and once with the new weights. The resulting KV values are then mixed with a learnable weight. These mixed values are then attended to during autoregressive generation.

They find this produces strong results on many datasets with many different LLMs. They also include a lot of ablation experiments.

**Strengths:**

The paper includes a lot of ablations that are very convincing of the efficacy of their approach. As I was reading, I keep thinking, I wonder if the see effect is because of X or Y instead of the bidirectional attention, and each time they had an ablation covering that possibility!

**Weaknesses:**

In Figure 2 they show that the initial value of the mixing ratio can cause large differences in the final average ratio once convergence is reached. Figure 3 shows some histograms of mixing ratios at different layers, and Table 6 has experiments that explore how the initial value affects performance. However, the scale of these experiments is small, and the results are inconclusive. Given the initial value causes such a large difference in the ratio, it would have been nice to see a deeper dive into this.

The experiments in the paper focus on datasets that have clear single-turn input/output pairs. Given that many LLMs these days are used for chat applications, it would have been nice to see some experiments on how bitune training effects these chat applications. As the text gets longs (more turns into the chat) does the effect of the initial bidirectional attention decrease? Does it help the model follow the initial prompt better? Can their approach be applied to mutli-turn dialogue (all of some of the user text is bidirectional which model generated text is causal?)

**Questions:**

Have you tried training the bitune attention matrices on a prefix language modeling task instead input/output formatted tasks? For example, randomly cutting a sequence of text in two pieces, applying bidirectional attention to the first part and then training on the second? This would also help answer the question of how much data is needed to learn the bitune matrices, which is mentioned but not answered in the paper.

---

> ### Author Response · Authors · 2024-11-22
>
> Thank you for your positive feedback and suggestions for new experiments.
>
> **Datasets beyond clear input-output pairs**
>
> We run additional experiments with Gemma-2B on the English subset of wikipedia [1] dataset, splitting each sequence at a random point, as suggested. We train the model for given number of steps (each step having 10 samples), and then evaluate it on the same benchmarks as in the Instruction Tuning setup, yielding the following results:
>
> | Model   | 500 steps  | 1000 steps | 3000 steps | 6000 stpes
> |---------|------|------|------|------|
> | Pretrained | 40.4 | 40.4| 40.4 | 40.4
> | LoRA    | 42.1 | 42.0 | 41.5  | 41.3
> | LoRA_16 | 41.9 | 41.7 | 41.9 | 41.5
> | Bitune  | **42.1** | **42.6** | **42.4**    | **42.5**
>
> The table shows average results for Bitune & baselines (base model, LoRA with rank 8, LoRA with rank 16), after a given number of update steps. Scores are averaged over benchmarks - PIQA, ARC, CSQA, SIQA, MMLU.
>
> We can see that Bitune leads to gains on downstream tasks even when trained on unstructured dataset, with improvements visible after 1000 update steps (10000 training samples).
>
> **Bitune for multi-turn dialogue**
>
> Regarding chat applications, the proposed method can be adapted to multi-turn dialogue settings in two ways:
>
> A) “_recompute_” The most straightforward way would be to treat the whole chat history as a prefix/query, and recompute the KV-cache for every new round - this would leverage bidirectional attention on all tokens, including the generated answers in previous rounds.
>
> B) “_alternating_” Another option is to have an alternating attention pattern - with the following procedure:
> 1. Prefill the first instruction and generate initial KV cache (by merging causal and bidirectional features)
> 2. Generate output token by token, appending KV of the output to the KV cache (here the KV of the output has only causal features)
> 3. Prefill the next instruction with two attention masks and append KV to the cache
> 4. Repeat from step 2
>
> [1] https://huggingface.co/datasets/wikimedia/wikipedia

---

### Official Review · Reviewer_xdHd · 2024-11-05

**Soundness:** 3
**Presentation:** 4
**Contribution:** 3
**Rating:** 6
**Confidence:** 4

**Summary:**

This paper aims to incorporate decoder-only language models with bidirectional attention. The authors introduce a second set of attention matrices for bidirectional understanding and integrate it with the original causal attention. Experimental results confirm the effectiveness of the proposed method, and comprehensive ablation studies offer insights into the design choices.

**Strengths:**

- The proposed method is simple but effective.
- Comprehensive ablation results with reasonable baselines.

**Weaknesses:**

Some existing works focus on prefix language models (non-causal decoder-only model) that utilize bidirectional attention for processing prefixes (i.e., inputs and instructions). Representative ones include U-PaLM [1] and UniLM [2]. How does the proposed method differ from these? While these models incorporate bidirectional attention during pre-training, the proposed method applies bidirectional attention specifically during instruction tuning. What would happen if we applied simple instruction tuning to these prefix language models (non-causal decoder-only model)? What are the advantages of bidirectional instruction tuning after causal decoder-only pre-training like the proposed method? I would like to see more discussions and even experiments on the comparison.

Another comment would be to visualize bidirectional attention on some examples. This could help highlight cases where bidirectional attention behaves differently from causal attention and therefore bring benefits.

Reference
1. Transcending Scaling Laws with 0.1% Extra Compute, 2022
2. Unified Language Model Pre-training for Natural Language Understanding and Generation, 2019

**Questions:**

Would the proposed bidirectional instruction tuning affect the quality for pure generation tasks?

---

> ### Author Response · Authors · 2024-11-22
>
> Thank you for your feedback - we would like to provide answers to your questions and points raised below.
>
> **How does Bitune compare to prior work?**
>
> >Some existing works focus on prefix language models that utilize bidirectional attention for processing prefixes. [...] How does the proposed method differ from these?
>
> The mentioned papers focus on pretraining, while our proposed method aims to effectively improve existing highly performant decoder-only models by finetuning them either on downstream tasks (Section 3.2), or on instruction-tuning datasets (Section 3.1). As we demonstrate in the paper, this can be done efficiently, even with academic compute.
>
> Moreover, the method itself differs - the most similar technique from mentioned papers is the “prefix-lm” objective/architecture, and it’s equivalent to our ablation variant called _naive bidir_ (Section 3.3). It applies bidirectional attention to the prefix/query while decoding the answer with causal attention. Our proposed method differs in the following ways:
>
> - We apply bidirectional *and* causal attention to the query, yielding two sets of features.
> - We use separate weights for obtaining these two sets of features.
> - We fuse both sets of features with learnable coefficients.
>
> With results in Table 3 we show that **the proposed method is superior to prefix-lm**/naive bidir. We will make this differentiation more clear in the final paper.
>
>
> **Bitune and text generation tasks**
>
> >Would the proposed bidirectional instruction tuning affect the quality for pure generation tasks?
>
> In the paper, we have tested Bitune on two kinds of benchmarks - Question Answering and Arithmetic Reasoning (GSM8K dataset). The latter involves open-ended generation before providing the final answer, and we see strong improvements on this benchmark across different model sizes, e.g. 9.3% improvement for Gemma-2B (Table 2), or 6.2% improvement for Codestral-22B (Table 14). This suggests that the proposed method indeed improves quality for _pure generation tasks_.
>
>
> **Visualization of bidirectional attention**
>
> >Another comment would be to visualize bidirectional attention on some examples. This could help highlight cases where bidirectional attention behaves differently from causal attention and therefore bring benefits.
>
> Thank you for this valuable suggestion. We have added two example pairs of attention matrix visualizations to the appendix (Section 6.9) of our updated paper, comparing bidirectional and causal attention patterns in the _Bituned_ Gemma-2B model. The attention matrices clearly demonstrate how the bidirectional component actively leverages information from future tokens, in contrast to the strictly sequential processing of causal attention.
>
> ---
>
> Thank you again for your feedback - we hope that our answer addresses your concerns and that you will consider raising your score.

---

> > ### Comment · Reviewer_xdHd · 2024-11-22
> >
> > Thanks for the response.
> >
> > Regarding prefix LM, I understand the difference. What I suggest is to include more discussion and results of prefix LM. For example, including prefix LM performance in Table 3 will bring more insights. Pre-trained prefix LM considers bidirectional attentions during pre-training, which can be treated as an upper-bound. If the proposed method has very close performance, that means considering bidirectional attentions during instruction tuning after decoder-only pre-training is enough, which would increase the value of this work. **The current naive bidir baseline is just an ablation, not really considering prefix LM as baselines.**

---

> > > ### Author Response · Authors · 2024-11-24
> > >
> > > We have evaluated the largest publicly available (to the best of our knowledge) decoder-only prefix-LM model - UniLM-Large, which has 340M parameters. Please see the results below, compared to the smallest decoder-only model we have tested.
> > >
> > >
> > > | Model | Method | PIQA | ARC | CSQA | SIQA | MMLU | Avg. |
> > > |-------|---------|------|-----|------|------|------|------|
> > > | UniLM-Large | Pretrained | 52.0 | 20.9 | 15.9 | 33.4 | 23.5 | 29.2 |
> > > | Gemma-2B | Pretrained | 57.5 | 36.9 | 35.5 | 38.2 | 34.0 | 40.4 |
> > > | | LoRA | 66.7 | 43.4 | 42.3 | 44.3 | 31.7 | 45.7 |
> > > | | LoRA₁₆ | 66.5 | 42.7 | 42.3 | 43.8 | 31.6 | 45.4 |
> > > | | Bitune | 69.6 | 47.5 | 46.9 | 49.5 | 35.3 | 49.7 |
> > >
> > > As demonstrated, UniLM significantly underperforms when compared to modern decoder-only LMs and cannot be used as a strict upper bound. For a fair comparison, we would need access to prefix-LM and causal models trained with identical data, parameters, and compute budgets, or pretrain them ourselves. However, this is out of scope for this work, and we leave it for future investigation.
> > >
> > > If you have any remaining questions, we're happy to clarify. Otherwise, having addressed all previous concerns, we welcome your reconsideration of the score.

---

> > > > ### Comment · Reviewer_xdHd · 2024-11-24
> > > >
> > > > Thanks for providing the results. Just wondering the UniLM is instruction tuned with the same instruction data or not.

---

> > > > > ### Author Response · Authors · 2024-11-25
> > > > >
> > > > > It's not, the above results are for the *base pretrained* UniLM-Large model, which can be compared to results of base Gemma-2B model.
> > > > >
> > > > > To give a complete picture, we conducted additional experiments where we fully finetuned UniLM on instruction-tuning data, *the same as used in our other experiments*. We performed hyperparameter search over different learning rates, and show the best obtained results below:
> > > > >
> > > > > | Model | Method | PIQA | ARC | CSQA | SIQA | MMLU | Avg. |
> > > > > |-------|---------|------|-----|------|------|------|------|
> > > > > | UniLM-Large | Pretrained | 52.0 | 20.9 | 15.9 | 33.4 | 23.5 | 29.2 |
> > > > > | | FullFT | 52.7 | 22.7 | 17.1 | 34.8 | 24.7 | 30.4 |
> > > > > | Gemma-2B | Pretrained | 57.5 | 36.9 | 35.5 | 38.2 | 34.0 | 40.4 |
> > > > > | | LoRA | 66.7 | 43.4 | 42.3 | 44.3 | 31.7 | 45.7 |
> > > > > | | LoRA₁₆ | 66.5 | 42.7 | 42.3 | 43.8 | 31.6 | 45.4 |
> > > > > | | Bitune | 69.6 | 47.5 | 46.9 | 49.5 | 35.3 | 49.7 |

---

> > > > > > ### Comment · Reviewer_xdHd · 2024-11-25
> > > > > >
> > > > > > Thanks for providing additional experimental results. I've increased my score to 6.

---

> ### Author Response · Authors · 2024-11-22
>
> Thank you for quick reply and this clarification. We agree that including pretrained prefix-LM models would indeed bring more insights, but unfortunately, for a fair comparison, there's no prefix-LM variants of the models we have tested available (trained with identical data, parameters, and compute budgets - differing only in their attention mechanism).
>
> We can include existing prefix-LM models like UniLM in our result tables, however _these will not serve as strict upper-bounds_ because of this. We will include these points in the discussion part of the updated work.

---

### Meta-Review · Area_Chair_x8Aw · 2024-12-19

**Metareview:**

This paper introduces a simple strategy to increase the representation capacity of transformer-based sequence models in situations where generation is to be made conditionally on some input context. Authors argue that in such a situation, similar to the now legacy encoder-decoder architecture, input contexts can be attended to bi-directionally, likely improving the representations via giving attention layers access to additional context.

As pointed out by some of the reviewers, this idea is not novel and models such as Prefix-LM operate similarly. I would also add that recent work has moved back to encoding context with bi-directional models (e.g., CEPE, XC-Cache, You Only Cache Once), though in those cases focus lied mostly on compressing the KV-cache rather than improving the representations. Authors replied arguing that their method has extra improvements over the basic approach in Prefix-LM, mostly related to the fact that they mix representations obtained bi-directionally and with a causal mask. In more detail, the proposed approach fine-tunes models with separate sets of parameters (adapters are used for parameter efficiency) for either causal or bi-directional attention over prefixes. During training, both representations are computed for the prefix part of the input and a convex combination is computed, with learned mixing parameters for each layer. Only the causal layers are used for the remainder of the input.

The idea is simple and results reported in the paper suggest it could be useful. The paper is clearly written and easy to follow. The paper received high scores, although acceptance wasn't unanimous among reviewers. I went through the paper and the discussion, and I have important concerns that prevent me from recommending acceptance. Please see below for a detailed discussion of these concerns.

- Lack of insight and justification of the design: the proposed approach isn't really well justified in the paper. I would expect some justification in the form of some of the following suggestions: theory showing the gap in representation capacity due to causal masks (i.e., moving to bi-directional strictly increases capacity), empirical evidence of the gap in representation capacity due to causal masks (i.e., some kind of simple experiment showing the effect of moving from one setting to the other), or references to work showing one of the previous results.

- Limited evaluation: the evaluation focuses on a single setting, instruction tuning, where instructions are treaded as prefixes. Testing is then carried out in different multiple-choice question answering tasks. There are plenty of settings that were typically tackled in the encoder-decoder era that could have been evaluated here. To name a few examples, it's unclear to what extent the proposal shows benefits in settings such as summarization and translation.

- Inconclusive evidence (main concern): results are reported in a rather inconclusive manner since rankings of models are performed based simply on average accuracy across a number of tasks. Differences in performances are small enough across models that concerns around significance of those differences must be raised. In table 2 for instance, I cannot conclude which method is the best performing one. Same goes for ablation results. 1-2% differences in average accuracy do not really suggest one approach is consistently better than another. Some form of variance would need to be accounted for if all the evidence presented is empirical and especially since authors claim significant improvements.

- The overhead the proposal incurs is not accounted for appropriately: the proposal effectively incurs a 2x overhead in compute and peak memory footprint for prefix encoding. While prefix encoding can be done in parallel to counter overhead in time to first token, that would still require at least twice as much compute. Reviewers brought that up in their reviews, and authors replied with results I believe downplay this limitation. Authors show TTFT results for a setting with a *very short* prefix, but give no insight in terms of how this scales. Memory overhead is not discussed up to a brief note in the limitations section.

**Additional Comments On Reviewer Discussion:**

Reviewers mostly raised concerns in terms of lack of novelty. To which authors replied detailing their novel contributions on top of the Prefix-LM approach. Some concerns were also raised in terms of the overhead the proposal incurs. The authors replied with some new extra results, but those focus on cases with a very short prefix, and more practical long context cases are not discussed.

---

### Decision · Program_Chairs · 2025-01-22

Reject